# Stochastic Neural Networks for Hierarchical Reinforcement Learning

**Carlos Florensa[†], Yan Duan[†‡], Pieter Abbeel[†‡§]**
[†] UC Berkeley, Department of Electrical Engineering and Computer Science
[§] International Computer Science Institute
[‡] OpenAI
florensa@berkeley.edu, {rocky,pieter}@openai.com

## ABSTRACT

Deep reinforcement learning has achieved many impressive results in recent years. However, tasks with sparse rewards or long horizons continue to pose significant challenges. To tackle these important problems, we propose a general framework that first learns useful skills in a pre-training environment, and then leverages the acquired skills for learning faster in downstream tasks. Our approach brings together some of the strengths of intrinsic motivation and hierarchical methods: the learning of useful skill is guided by a single proxy reward, the design of which requires very minimal domain knowledge about the downstream tasks. Then a high-level policy is trained on top of these skills, providing a significant improvement of the exploration and allowing to tackle sparse rewards in the downstream tasks. To efficiently pre-train a large span of skills, we use Stochastic Neural Networks combined with an information-theoretic regularizer. Our experiments[1] show[2] that this combination is effective in learning a wide span of interpretable skills in a sample-efficient way, and can significantly boost the learning performance uniformly across a wide range of downstream tasks.

## 1 INTRODUCTION

In recent years, deep reinforcement learning has achieved many impressive results, including playing Atari games from raw pixel inputs (Guo et al., 2014; Mnih et al., 2015; Schulman et al., 2015), mastering the game of Go (Silver et al., 2016), and acquiring advanced manipulation and locomotion skills from raw sensory inputs (Schulman et al., 2015; Lillicrap et al., 2015; Watter et al., 2015; Schulman et al., 2016; Heess et al., 2015; Levine et al., 2016). Despite these success stories, these deep RL algorithms typically employ naive exploration strategies such as $\epsilon$-greedy or uniform Gaussian exploration noise, which have been shown to perform poorly in tasks with sparse rewards (Duan et al., 2016; Houthooft et al., 2016; Bellemare et al., 2016). Tasks with sparse rewards are common in realistic scenarios. For example, in navigation tasks, the agent is not rewarded until it finds the target. The challenge is further complicated by long horizons, where naive exploration strategies can lead to exponentially large sample complexity (Osband et al., 2014).

To tackle these challenges, two main strategies have been pursued: The first strategy is to design a hierarchy over the actions (Parr & Russell, 1998; Sutton et al., 1999; Dietterich, 2000). By composing low-level actions into high-level primitives, the search space can be reduced exponentially. However, these approaches require domain-specific knowledge and careful hand-engineering. The second strategy uses intrinsic rewards to guide exploration (Schmidhuber, 1991; 2010; Houthooft et al., 2016; Bellemare et al., 2016). The computation of these intrinsic rewards does not require domain-specific knowledge. However, when facing a collection of tasks, these methods do not provide a direct answer about how knowledge about solving one task may transfer to other tasks, and by solving each of the tasks from scratch, the overall sample complexity may still be high.

---

[1]Code available at: https://github.com/florensacc/snn4hrl
[2]Videos available at: http://bit.ly/snn4hrl-videos

In this paper, we propose a general framework for training policies for a collection of tasks with sparse rewards. Our framework first learns a span of skills in a pre-training environment, where it is employed with nothing but a proxy reward signal, whose design only requires very minimal domain knowledge about the downstream tasks. This proxy reward can be understood as a form of intrinsic motivation that encourages the agent to explore its own capabilities, without any goal information or sensor readings specific to each downstream task. The set of skills can be used later in a wide collection of different tasks, by training a separate high-level policy for each task on top of the skills, thus reducing sample complexity uniformly. To learn the span of skills, we propose to use Stochastic Neural Networks (SNNs) (Neal, 1990; 1992; Tang & Salakhutdinov, 2013), a class of neural networks with stochastic units in the computation graph. This class of architectures can easily represent multi-modal policies, while achieving weight sharing among the different modes. We parametrize the stochasticity in the network by feeding latent variables with simple distributions as extra input to the policy. In the experiments, we observed that direct application of SNNs does not always guarantee that a wide range of skills will be learned. Hence, we propose an information-theoretic regularizer based on Mutual Information (MI) in the pre-training phase to encourage diversity of behaviors of the SNN policy. Our experiments find that our hierarchical policy-learning framework can learn a wide range of skills, which are clearly interpretable as the latent code is varied. Furthermore, we show that training high-level policies on top of the learned skills can results in strong performance on a set of challenging tasks with long horizons and sparse rewards.

## 2 RELATED WORK

One of the main appealing aspects of hierarchical reinforcement learning (HRL) is to use skills to reduce the search complexity of the problem (Parr & Russell, 1998; Sutton et al., 1999; Dietterich, 2000). However, specifying a good hierarchy by hand requires domain-specific knowledge and careful engineering, hence motivating the need for learning skills automatically. Prior work on automatic learning of skills has largely focused on learning skills in discrete domains (Chentanez et al., 2004; Vigorito & Barto, 2010). A popular approach there is to use statistics about state transitions to identify bottleneck states (Stolle & Precup, 2002; Mannor et al., 2004; Şimşek et al., 2005). It is however not clear how these techniques can be applied to settings with high-dimensional continuous spaces. More recently, Mnih et al. (2016) propose a DRAW-like (Gregor et al., 2015) recurrent neural network architecture that can learn temporally extended macro actions. However, the learning needs to be guided by external rewards as supervisory signals, and hence the algorithm cannot be straightforwardly applied in sparse reward settings.

There has also been work on learning skills in tasks with continuous actions (Schaal et al., 2005; Konidaris et al., 2011; Daniel et al., 2013; Ranchod et al., 2015). These methods extract useful skills from successful trajectories for the same (or closely related) tasks, and hence require first solving a comparably challenging task or demonstrations. Guided Policy Search (GPS) Levine et al. (2016) leverages access to training in a simpler setting. GPS trains with iLQG Todorov & Li (2005) in state space, and in parallel trains a neural net that only receives raw sensory inputs that has to agree with the iLQG controller. The neural net policy is then able to generalize to new situations.

Another line of work on skill discovery particularly relevant to our approach is the HiREPS algorithm by Daniel et al. (2012) where, instead of having a mutual information bonus like in our proposed approach, they introduce a constraint on an equivalent metric. The solution approach is nevertheless very different as they cannot use policy gradients. Furthermore, although they do achieve multimodality like us, they only tried the episodic case where a single option is active during the rollout. Hence the hierarchical use of the learned skills is less clear. Similarly, the Option-critic architecture (Bacon & Precup, 2016) can learn interpretable skills, but whether they can be reuse across complex tasks is still an open question. Recently, Heess et al. (2016) have independently proposed to learn a range of skills in a pre-training environment that will be useful for the downstream tasks, which is similar to our framework. However, their pre-training setup requires a set of goals to be specified. In comparison, we use a signle proxy reward as the only signal to the agent during the pre-training phase, the construction of which only requires minimal domain knowledge or instrumentation of the environment.

## 3 Preliminaries

We define a discrete-time finite-horizon discounted Markov decision process (MDP) by a tuple $M = (\mathcal{S}, \mathcal{A}, \mathcal{P}, r, \rho_0, \gamma, T)$, in which $\mathcal{S}$ is a state set, $\mathcal{A}$ an action set, $\mathcal{P} : \mathcal{S} \times \mathcal{A} \times \mathcal{S} \to \mathbb{R}_+$ a transition probability distribution, $r : \mathcal{S} \times \mathcal{A} \to [-R_{\max}, R_{\max}]$ a bounded reward function, $\rho_0 : \mathcal{S} \to \mathbb{R}_+$ an initial state distribution, $\gamma \in [0, 1]$ a discount factor, and $T$ the horizon. When necessary, we attach a suffix to the symbols to resolve ambiguity, such as $\mathcal{S}^M$. In policy search methods, we typically optimize a stochastic policy $\pi_\theta : \mathcal{S} \times \mathcal{A} \to \mathbb{R}_+$ parametrized by $\theta$. The objective is to maximize its expected discounted return, $\eta(\pi_\theta) = \mathbb{E}_\tau[\sum_{t=0}^{T} \gamma^t r(s_t, a_t)]$, where $\tau = (s_0, a_0, \ldots)$ denotes the whole trajectory, $s_0 \sim \rho_0(s_0)$, $a_t \sim \pi_\theta(a_t|s_t)$, and $s_{t+1} \sim \mathcal{P}(s_{t+1}|s_t, a_t)$.

## 4 Problem statement

Our main interest is in solving a collection of downstream tasks, specified via a collection of MDPs $\mathcal{M}$. If these tasks do not share any common structure, we cannot expect to acquire a set of skills that will speed up learning for all of them. On the other hand, we want the structural assumptions to be minimal, to make our problem statement more generally applicable.

For each MDP $M \in \mathcal{M}$, we assume that the state space $\mathcal{S}^M$ can be factored into two components, $\mathcal{S}_{\text{agent}}$, and $\mathcal{S}_{\text{rest}}^M$, which only weakly interact with each other. The $\mathcal{S}_{\text{agent}}$ should be the same for all MDPs in $\mathcal{M}$. We also assume that all the MDPs share the same action space. Intuitively, we consider a robot who faces a collection of tasks, where the dynamics of the robot are shared across tasks, and are covered by $\mathcal{S}_{\text{agent}}$, but there may be other components in a task-specific state space, which will be denoted by $\mathcal{S}_{\text{rest}}$. For instance, in a grasping task $M$, $\mathcal{S}_{\text{rest}}^M$ may include the positions of objects at interest or the new sensory input associated with the task. This specific structural assumption has been studied in the past as sharing the same *agent-space* (Konidaris & Barto, 2007).

Given a collection of tasks satisfying the structural assumption, our objective for designing the algorithm is to minimize the total sample complexity required to solve these tasks. This has been more commonly studied in the past in sequential settings, where the agent is exposed to a sequence of tasks, and should learn to make use of experience gathered from solving earlier tasks to help solve later tasks (Taylor & Stone, 2009; Wilson et al., 2007; Lazaric & Ghavamzadeh, 2010; Devin et al., 2017). However, this requires that the earlier tasks are relatively easy to solve (e.g., through good reward shaping or simply being easier) and is not directly applicable when all the tasks have sparse rewards, which is a much more challenging setting. In the next section, we describe our formulation, which takes advantage of a pre-training task that can be constructed with minimal domain knowledge, and can be applied to the more challenging scenario.

## 5 Methodology

In this section, we describe our formulation to solve a collection of tasks, exploiting the structural assumption articulated above. In Sec. 5.1, we describe the pre-training environment, where we use proxy rewards to learn a useful span of skills. In Sec. 5.2, we motivate the usage of Stochastic Neural Networks (SNNs) and discuss the architectural design choices made to tailor them to skill learning for RL. In Sec. 5.3, we describe an information-theoretic regularizer that further improves the span of skills learned by SNNs. In Sec. 5.4, we describe the architecture of high-level policies over the learned skills, and the training procedure for the downstream tasks with sparse rewards. Finally, in Sec. 5.5, we describe the policy optimization for both phases of training.

### 5.1 Constructing the pre-training environment

Given a collection of tasks, we would like to construct a pre-training environment, where the agent can learn a span of skills, useful for enhancing exploration in downstream tasks. We achieve this by letting the agent freely interact with the environment in a minimal setup. For a mobile robot, this can be a spacious environment where the robot can first learn the necessary locomotion skills; for a manipulator arm which will be used for object manipulation tasks, this can be an environment with many objects that the robot can interact with.

The skills learned in this environment will depend on the reward given to the agent. Rather than setting different rewards in the pre-training environment corresponding to the desired skills, which requires precise specification about what each skill should entail, we use a generic proxy reward as the only reward signal to guide skill learning. The design of the proxy reward should encourage the existence of locally optimal solutions, which will correspond to different skills the agent should learn. In other words, it encodes the prior knowledge about what high level behaviors might be useful in the downstream tasks, rewarding all of them roughly equally. For a mobile robot, this reward can be as simple as proportional to the magnitude of the speed of the robot, without constraining the direction of movement. For a manipulator arm, this reward can be the successful grasping of any object.

Every time we train a usual uni-modal gaussian policy in such environment, it should converge towards a potentially different skill. As seen in Sec. 7.1, applying this approach to mobile robots gives a different direction (and type) of locomotion every time. But it is an inefficient procedure: training each policy from scratch is not sample efficient (sample complexity grows linearly with the number of skills to be learned) and nothing encourages the individual skills to be distinct. The first issue will be addressed in the next subsection by using Stochastic Neural Networks as policies and the second issue will be addressed in Sec. 5.3 by adding an information-theoretic regularizer.

## 5.2    STOCHASTIC NEURAL NETWORKS FOR SKILL LEARNING

To learn several skills at the same time, we propose to use Stochastic Neural Networks (SNNs), a general class of neural networks with stochastic units in the computation graph. There has been a large body of prior work on special classes of SNNs, such as Restricted Boltzmann Machines (RBMs) (Smolensky, 1986; Hinton, 2002), Deep Belief Networks (DBNs) (Hinton et al., 2006), and Sigmoid Belief Networks (SBNs) (Neal, 1990; Tang & Salakhutdinov, 2013). They have rich representation power and can in fact approximate any well-behaved probability distributions (Le Roux & Bengio, 2008; Cho et al., 2013). Policies modeled via SNNs can hence represent complex action distributions, especially multi-modal distributions.

For our purpose, we use a simple class of SNNs, where latent variables with fixed distributions are integrated with the inputs to the neural network (here, the observations from the environment) to form a joint embedding, which is then fed to a standard feed-forward neural network (FNN) with deterministic units, that computes distribution parameters for a uni-modal distribution (e.g. the mean and variance parameters of a multivariate Gaussian). We use simple categorical distributions with uniform weights for the latent variables, where the number of classes, $K$, is an hyperparameter that upper bounds the number of skills that we would like to learn.

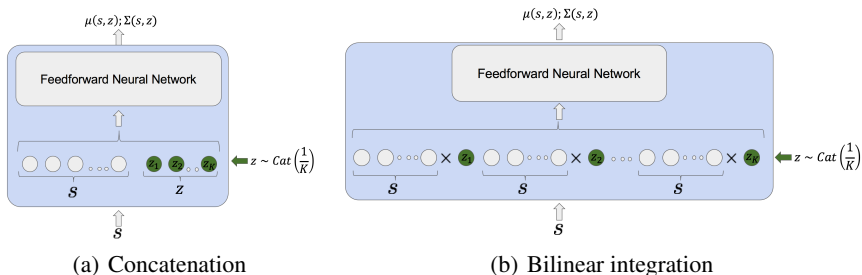

(a) Concatenation                    (b) Bilinear integration

Figure 1: Different architectures for the integration of the latent variables in a FNN

The simplest joint embedding, as shown in Figure 1(a), is to concatenate the observations and the latent variables directly.[3] However, this limits the expressiveness power of integration between the observation and latent variable. Richer forms of integrations, such as multiplicative integrations and bilinear pooling, have been shown to have greater representation power and improve the optimization landscape, achieving better results when complex interactions are needed (Fukui et al., 2016; Wu et al., 2016). Inspired by this work, we study using a simple bilinear integration, by forming the outer product between the observation and the latent variable (Fig. 1(b)). Note that the concatenation

---

[3]In scenarios where the observation is very high-dimensional such as images, we can form a low-dimensional embedding of the observation alone, say using a neural network, before jointly embedding it with the latent variable.

integration effectively corresponds to changing the bias term of the first hidden layer depending on the latent code $z$ sampled, and the bilinear integration to changing all the first hidden layer weights. As shown in the experiments, the choice of integration greatly affects the quality of the span of skills that is learned. Our bilinear integration already yields a large span of skills, hence no other type of SNNs is studied in this work.

To obtain temporally extended and consistent behaviors associated with each latent code, in the pre-training environment we sample a latent code at the beginning of every rollout and keep it constant throughout the entire rollout. After training, each of the latent codes in the categorical distribution will correspond to a different, interpretable skill, which can then be used for the downstream tasks.

Compared to training separate policies, training a single SNN allows for flexible weight-sharing schemes among different policies. It also takes a comparable amount of samples to train an SNN than a regular uni-modal gaussian policy, so the sample efficiency of training $K$ skills is effectively $K$ times better. To further encourage the diversity of skills learned by the SNN we introduce an information theoretic regularizer, as detailed in the next section.

### 5.3 INFORMATION-THEORETIC REGULARIZATION

Although SNNs have sufficient expressiveness to represent multi-modal policies, there is nothing in the optimization that prevents them from collapsing into a single mode. We have not observed this worst case scenario in our experiments, but sometimes different latent codes correspond to very similar skills. It is desirable to have direct control over the diversity of skills that will be learned. To achieve this, we introduce an information-theoretic regularizer, inspired by recent success of similar objectives in encouraging interpretable representation learning in InfoGAN Chen et al. (2016).

Concretely, we add an additional reward bonus, proportional to the mutual information (MI) between the latent variable and the current state. We only measure the MI with respect to a relevant subset of the state. For a mobile robot, we choose this to be the $c = (x, y)$ coordinates of its center of mass (CoM). Formally, let $C$ be a random variable denoting the current CoM coordinate of the agent, and let $Z$ be the latent variable. Then the MI can be expressed as $I(Z; C) = H(Z) - H(Z|C)$, where $H$ denotes the entropy function. In our case, $H(Z)$ is constant since the probability distribution of the latent variable is fixed to uniform during this stage of training. Hence, maximizing the MI is equivalent to minimizing the conditional entropy $H(Z|C)$. As entropy is a measure of uncertainty, another interpretation of this bonus is that, given where the robot is, it should be easy to infer which skill the robot is currently performing. To penalize $H(Z|C) = -\mathbb{E}_{z,c} \log p(Z = z|C = c)$, we modify the reward received at every step as specified in Eq. (1), where $\hat{p}(Z = z^n|c_t^n)$ is an estimate of the posterior probability of the latent code $z^n$ sampled on rollout $n$, given the coordinates $c_t^n$ at time $t$ of that rollout.

$$R_t^n \leftarrow R_t^n + \alpha_H \log \hat{p}(Z = z^n|c_t^n) \tag{1}$$

To estimate the posterior $\hat{p}(Z = z^n|c_t^n)$, we apply the following discretization: we partition the $(x, y)$ coordinate space into cells and map the continuous-valued CoM coordinates into the cell containing it. Overloading notation, now $c_t$ is a discrete variable indicating the cell where the CoM is at time $t$. This way, calculation of the empirical posterior only requires maintaining visitation counts $m_c(z)$ of how many times each cell $c$ is visited when latent code $z$ is sampled. Given that we use a batch policy optimization method, we use all trajectories of the current batch to compute all the $m_c(z)$ and estimate $\hat{p}(Z = z^n|c_t^n)$, as shown in Eq. (2). If $c$ is more high dimensional, the posterior $\hat{p}(Z = z^n|c_t^n)$ can also be estimated by fitting a MLP regressor by Maximum Likelihood.

$$\hat{p}(Z = z|(x, y)) \approx \hat{p}(Z = z|c) = \frac{m_c(z)}{\sum_{z'} m_c(z')} \tag{2}$$

### 5.4 LEARNING HIGH-LEVEL POLICIES

Given a span of $K$ skills learned during the pre-training task, we now describe how to use them as basic building blocks for solving tasks where only sparse reward signals are provided. Instead of learning from scratch the low-level controls, we leverage the provided skills by freezing them and training a high-level policy (Manager Neural Network) that operates by selecting a skill and committing to it for a fixed amount of steps $\mathcal{T}$. The imposed temporal consistency and the quality

of the skills (in our case given by optimizing the proxy reward in the pre-train environment) yield an enhanced exploration allowing to solve the downstream tasks with sparse rewards.

For any given task $M \in \mathcal{M}$ we train a new Manager NN on top of the common skills. Given the factored representation of the state space $\mathcal{S}^M$ as $\mathcal{S}_{\text{agent}}$ and $\mathcal{S}_{\text{rest}}^M$, the high-level policy receives the full state as input, and outputs the parametrization of a categorical distribution from which we sample a discrete action $z$ out of $K$ possible choices, corresponding to the $K$ available skills. If those skills are independently trained uni-modal policies, $z$ dictates the policy to use during the following $\mathcal{T}$ time-steps. If the skills are encapsulated in a SNN, $z$ is used in place of the latent variable. The architecture is depicted in Fig. 2.

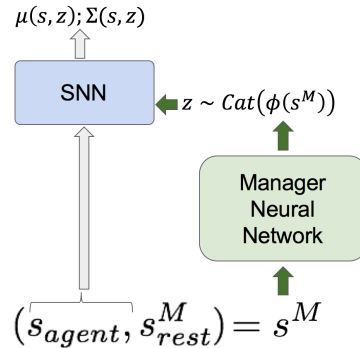

The weights of the low level and high level neural networks could also be jointly optimized to adapt the skills to the task at hand. This end-to-end training of a policy with discrete latent variables in the Stochastic Computation Graph could be done using straight-through estimators like the one proposed by Jang et al. (2017) or Maddison et al. (2017). Nevertheless, we show in our experiments that frozen low-level policies are already sufficient to achieve good performance in the studied downstream tasks, so these directions are left as future research.

Figure 2: Hierarchical SNN architecture to solve downstream tasks

## 5.5 Policy Optimization

For both the pre-training phase and the training of the high-level policies, we use Trust Region Policy Optimization (TRPO) as the policy optimization algorithm (Schulman et al., 2015). We choose TRPO due to its excellent empirical performance and because it does not require excessive hyperparameter tuning. The training on downstream tasks does not require any modifications, except that the action space is now the set of skills that can be used. For the pre-training phase, due to the presence of categorical latent variables, the marginal distribution of $\pi(a|s)$ is now a mixture of Gaussians instead of a simple Gaussian, and it may even become intractable to compute if we use more complex latent variables. To avoid this issue, we consider the latent code as part of the observation. Given that $\pi(a|s, z)$ is still a Gaussian, TRPO can be applied without any modification.

---

**Algorithm 1:** Skill training for SNNs with MI bonus

---

**Initialize:** Policy $\pi_\theta$; Latent dimension $K$;
**while** *Not trained* **do**
 **for** $n \leftarrow 1$ **to** $N$ **do**
 Sample $z_n \sim \text{Cat}\left(\frac{1}{K}\right)$;
 Collect rollout with $z_n$ fixed;
 **end**
 Compute $\hat{p}(Z = z|c) = \frac{m_c(z)}{\sum_{z'} m_c(z')}$;
 Modify $R_t^n \leftarrow R_t^n + \alpha_H \log \hat{p}(Z = z^n|c_t^n)$;
 Apply TRPO considering $z$ part of the observation;
**end**

---

## 6 Experiments

We have applied our framework to the two hierarchical tasks described in the benchmark by Duan et al. (2016): Locomotion + Maze and Locomotion + Food Collection (Gather). The observation space of these tasks naturally decompose into $S_{agent}$ being the robot and $S_{rest}^M$ the task-specific attributes like walls, goals, and sensor readings. Here we report the results using the Swimmer robot, also described in the benchmark paper. In fact, the swimmer locomotion task described therein corresponds exactly to our pretrain task, as we also solely reward speed in a plain environment. We report results with more complex robots in Appendix C-D.

To increase the variety of downstream tasks, we have constructed four different mazes. Maze 0 is the same as the one described in the benchmark (Duan et al., 2016) and Maze 1 is its reflection, where the robot has to go backwards-right-right instead of forward-left-left. These correspond to Figs. 3(a)-3(b), where the robot is shown in its starting position and has to reach the other end of the U turn. Mazes 2 and 3 are different instantiations of the environment shown in Fig. 3(c), where the goal has been placed in the North-East or in the South-West corner respectively. A reward of 1 is only granted when the robot reaches the goal position. In the Gather task depicted in Fig. 3(d), the robot gets a reward of 1 for collecting green balls and a reward of -1 for the red ones, all of which are positioned randomly at the beginning of each episode. Apart from the robot joints positions and velocities, in these tasks the agent also receives LIDAR-like sensor readings of the distance to walls, goals or balls that are within a certain range.

In the benchmark of continuous control problems (Duan et al., 2016) it was shown that algorithms that employ naive exploration strategies could not solve them. More advanced intrinsically motivated explorations (Houthooft et al., 2016) do achieve some progress, and we report our stronger results with the exact same setting in Appendix B. Our hyperparameters for the neural network architectures and algorithms are detailed in the Appendix A and the full code is available[4].

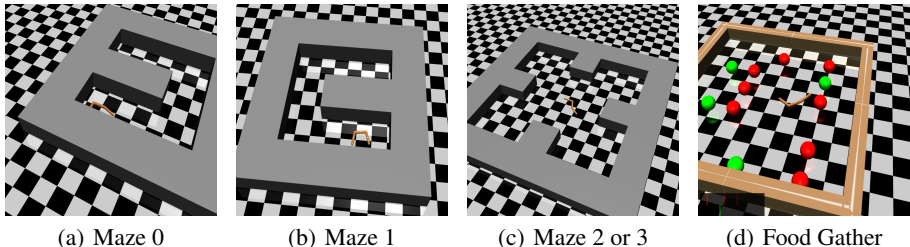

|       (a) Maze 0       |       (b) Maze 1       |     (c) Maze 2 or 3     |     (d) Food Gather     |

Figure 3: Illustration of the sparse reward tasks

# 7 RESULTS

We evaluate every step of the skill learning process, showing the relevance of the different pieces of our architecture and how they impact the exploration achieved when using them in a hierarchical fashion. Then we report the results[5] on the sparse environments described above. We seek to answer the following questions:

- Can the multimodality of SNNs and the MI bonus consistently yield a large span of skills?
- Can the pre-training experience improve the exploration in downstream environments?
- Does the enhanced exploration help to efficiently solve sparse complex tasks?

## 7.1 SKILL LEARNING IN PRETRAIN

To evaluate the diversity of the learned skills we use "visitation plots", showing the $(x, y)$ position of the robot's Center of Mass (CoM) during 100 rollouts of 500 time-steps each. At the beginning of every rollout, we reset the robot to the origin, always in the same orientation (as done during training). In Fig. 4(a) we show the visitation plot of six different feed-forward policies, each trained from scratch in our pre-training environment. For better graphical interpretation and comparison with the next plots of the SNN policies, Fig. 4(b) superposes a batch of 50 rollout for each of the 6 policies, each with a different color. Given the morphology of the swimmer, it has a natural preference for forward and backward motion. Therefore, when no extra incentive is added, the visitation concentrates heavily on the direction it is always initialized with. Note nevertheless that the proxy reward is general enough so that each independently trained policy yields a different way of advancing, hence granting a potentially useful skills to solve downstream tasks when embedded in our described Multi-policy hierarchical architecture.

Next we show that, using SNNs, we can learn a similar or even larger span of skills without training several independent policies. The number of samples used to train a SNN is the same as a single

---

[4]Code available at: `https://github.com/florensacc/snn4hrl`
[5]Videos available at: `http://bit.ly/snn4hrl-videos`

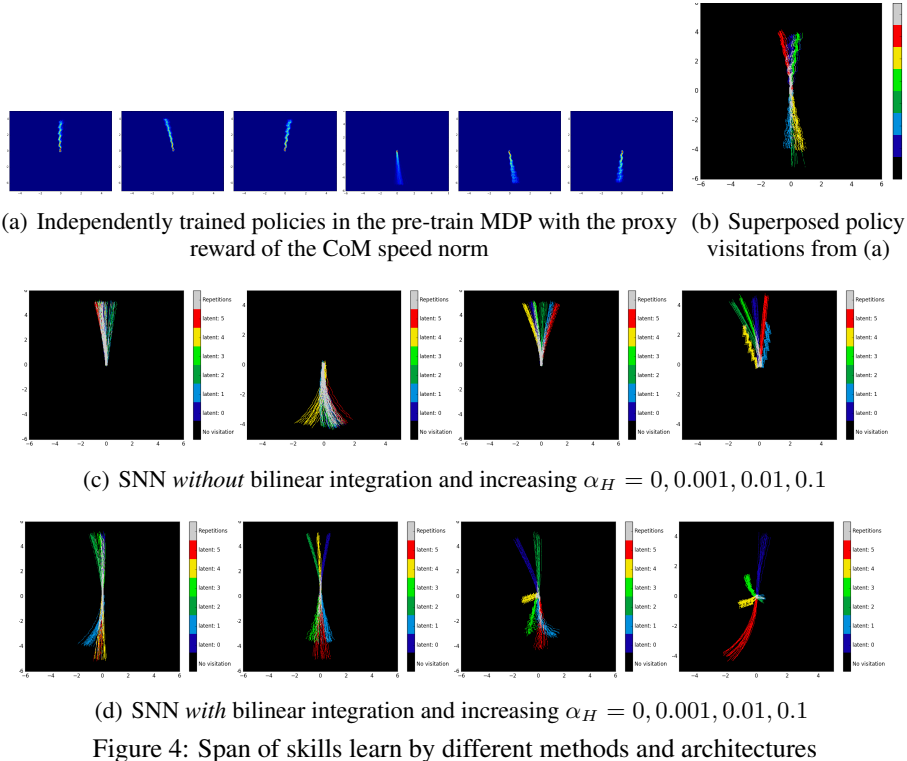

(a) Independently trained policies in the pre-train MDP with the proxy reward of the CoM speed norm

(b) Superposed policy visitations from (a)

(c) SNN *without* bilinear integration and increasing $\alpha_H = 0, 0.001, 0.01, 0.1$

(d) SNN *with* bilinear integration and increasing $\alpha_H = 0, 0.001, 0.01, 0.1$

Figure 4: Span of skills learn by different methods and architectures

feed-forward policy from Fig. 4(a), therefore this method of learning skills effectively reduces the sample complexity by a factor equal to the numbers of skills being learned. With an adequate architecture and MI bonus, we show that the span of skills generated is also richer. In the two rows of Figs. 4(c)-4(d) we present the visitation plots of SNNs policies obtained for different design choices. Now the colors indicate the latent code that was sampled at the beginning of the rollout. We observe that each latent code generates a particular, interpretable behavior. Given that the initialization orientation is always the same, the different skills are truly distinct ways of moving: forward, backwards, or sideways. In the following we analyze the impact on the span of skills of the integration of latent variables (concatenation in first row and bilinear in the second) and the MI bonus (increasing coefficient $\alpha_H$ towards the right). Simple concatenation of latent variables with the observations rarely yields distinctive behaviors for each latent code sampled. On the other hand, we have observed that 80% of the trained SNNs with bilinear integration acquire at least forward and backward motion associated with different latent codes. This can be further improved to 100% by increasing $\alpha_H$ and we also observe that the MI bonus yields less overlapped skills and new advancing/turning skills, independently of the integration of the latent variables.

## 7.2 HIERARCHICAL USE OF SKILLS

The hierarchical architectures we propose have a direct impact on the areas covered by random exploration. We will illustrate it with plots showing the visitation of the $(x - y)$ position of the Center of Mass (CoM) during single rollouts of one million steps, each generated with a different architecture.

On one hand, we show in Fig. 5(a) the exploration obtained with actions drawn from a Gaussian with $\mu = 0$ and $\Sigma = I$, similar to what would happen in the first iteration of training a Multi-Layer Perceptron with normalized random initialization of the weights. This noise is relatively large as the swimmer robot has actions clipped to $[-1, 1]$. Still, it does not yield good exploration as all point reached by the robot during the one million steps rollout lay within the $[-2, 2] \times [-2, 2]$ box around the initial position. This exploration is not enough to reach the first reward in most of the downstream sparse reward environments and will never learn, as already reported by Duan et al. (2016).

On the other hand, using our hierarchical structures with pretrained policies introduced in Sec. 5.4 yields a considerable increase in exploration, as reported in Fig. 5(b)-5(d). These plots also show a possible one millions steps rollout, but under a randomly initialized Manager Network, hence outputting uniformly distributed one-hot vectors every $\mathcal{T} = 500$ steps. The color of every point in the CoM trajectory corresponds to the latent code that was fixed at that time-step, clearly showing the "skill changes". In the following we describe what specific architecture was used for each plot.

The rollout in Fig. 5(b) is generated following a policy with the *Multi-policy* architecture. This hierarchical architecture is our strong baseline given that it has used six times more samples for pretrain because it trained six policies independently. As explained in Sec. 5.4, every $\mathcal{T} = 500$ steps it uses the sampled one-hot output of the Manager Network to select one of the six pre-trained policies. We observe that the exploration given by the *Multi-policy* hierarchy heavily concentrates around upward and downward motion, as is expected from the six individual pre-trained policies composing it (refer back to Fig. 4(a)).

Finally, the rollouts in Fig.5(c) and 5(d) use our hierarchical architecture with a SNN with bilinear integration and $\alpha_H = 0$ and 0.01 respectively. As described in Sec. 5.4, the placeholder that during the SNN training received the categorical latent code now receives the one-hot vector output of the Manager Network instead. The exploration obtained with SNNs yields a wider coverage of the space as the underlying policy usually has a larger span of skills. Note how most of the changes of latent code yield a corresponding change in direction. Our simple pre-train setup and the interpretability of the obtained skills are key advantages with respect to previous hierarchical structures.

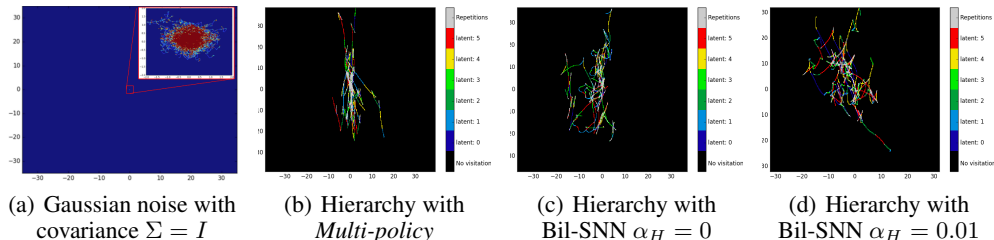

(a) Gaussian noise with covariance $\Sigma = I$ (b) Hierarchy with *Multi-policy* (c) Hierarchy with Bil-SNN $\alpha_H = 0$ (d) Hierarchy with Bil-SNN $\alpha_H = 0.01$

Figure 5: Visitation plots for different randomly initialized architectures (one rollout of 1M steps). All axis are in scale [-30,30] and we added a zoom for the Gaussian noise to scale [-2,2]

### 7.3 MAZES AND GATHER TASKS

In Fig. 6 we evaluate the learning curves of the different hierarchical architectures proposed. Due to the sparsity of these tasks, none can be properly solved by standard reinforcement algorithms (Duan et al., 2016). Therefore, we compare our methods against a better baseline: adding to the downstream task the same Center of Mass (CoM) proxy reward that was granted to the robot in the pre-training task. This baseline performs quite poorly in all the mazes, Fig. 6(a)-6(c). This is due to the long time-horizon needed to reach the goal and the associated credit assignment problem. Furthermore, the proxy reward alone does not encourage diversity of actions as the MI bonus for SNN does. The proposed hierarchical architectures are able to learn much faster in every new MDP as they effectively shrink the time-horizon by aggregating time-steps into useful primitives. The benefit of using SNNs with bilinear integration in the hierarchy is also clear over most mazes, although pretraining with MI bonus does not always boost performance. Observing the learned trajectories that solve the mazes, we realize that the turning or sideway motions (more present in SNNs pretrained with MI bonus) help on maze 0, but are not critical in these tasks because the robot may use a specific forward motion against the wall of the maze to reorient itself. We think that the extra skills will shine more in other tasks requiring more precise navigation. And indeed this is the case in the Gather task as seen in Fig. 6(d), where not only the average return increases, but also the variance of the learning curve is lower for the algorithm using SNN pretrained with MI bonus, denoting a more consistent learning across different SNNs. For more details on the Gather task, refer to Appendix B.

It is important to mention that each curve of SNN or Multi-policy performance corresponds to a total of 10 runs: 2 random seeds for 5 different SNNs obtained with 5 random seeds in the pretrain task. There is no cherry picking of the pre-trained policy to use in the sparse environment, as done

in most previous work. This also explains the high variance of the proposed hierarchical methods in some tasks. This is particularly hard on the mazes as some pretrained SNN may not have the particular motion that allows to reach the goal, so they will have a 0 reward. See Appendix C.4 for more details.

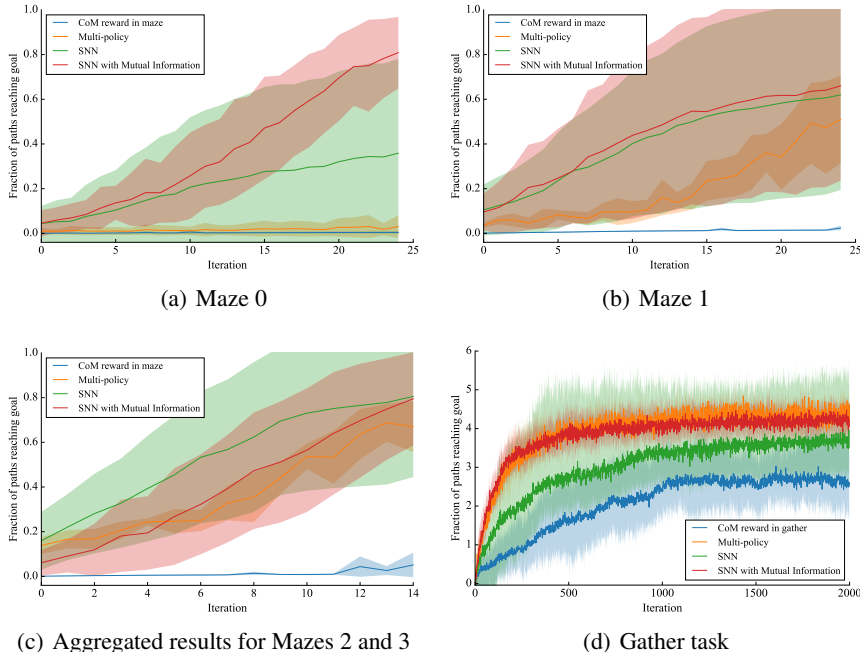

(a) Maze 0

(b) Maze 1

(c) Aggregated results for Mazes 2 and 3

(d) Gather task

Figure 6: Faster learning of the hierarchical architectures in the sparse downstream MDPs

## 8 DISCUSSION AND FUTURE WORK

We propose a framework for first learning a diverse set of skills using Stochastic Neural Networks trained with minimum supervision, and then utilizing these skills in a hierarchical architecture to solve challenging tasks with sparse rewards. Our framework successfully combines two parts, firstly an unsupervised procedure to learn a large span of skills using proxy rewards and secondly a hierarchical structure that encapsulates the latter span of skills and allows to re-use them in future tasks. The span of skills learning can be greatly improved by using Stochastic Neural Networks as policies and their additional expressiveness and multimodality. The bilinear integration and the mutual information bonus are key to consistently yield a wide, interpretable span of skills. As for the hierarchical structure, our experiments demonstrate it can significantly boost the exploration of an agent in a new environment and we demonstrate its relevance for solving complex tasks as mazes or gathering.

One limitation of our current approach is the switching between skills for unstable agents, as reported in the Appendix D.2 for the "Ant" robot. There are multiple future directions to make our framework more robust to such challenging robots, like learning a transition policy or integrating switching in the pretrain task. Other limitations of our current approach are having fixed sub-policies and a fixed switch time $\mathcal{T}$ during the training of downstream tasks. The first issue can be alleviated by introducing end-to-end training, for example using the new straight-through gradient estimators for Stochastic Computations Graphs with discrete latent variables (Jang et al., 2017; Maddison et al., 2017). The second issue is not critical for static tasks like the ones used here, as studied in Appendix C.3. But in case of becoming a bottleneck for more complex dynamic tasks, a termination policy could be learned by the Manager, similar to the option framework. Finally, we only used feedforward architectures and hence the decision of what skill to use next only depends on the observation at the moment of switching, not using any sensory information gathered while the previous skill was active. This limitation could be eliminated by introducing a recurrent architecture at the Manager level.

ACKNOWLEDGMENTS

This work was supported in part by DARPA, by the Berkeley Vision and Learning Center (BVLC), by the Berkeley Artificial Intelligence Research (BAIR) laboratory, by Berkeley Deep Drive (BDD), and by ONR through a PECASE award. Carlos Florensa was also supported by a La Caixa Ph.D. Fellowship, and Yan Duan by a BAIR Fellowship and a Huawei Fellowship.

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

## A HYPERPARAMETERS

All policies are trained with TRPO with step size $0.01$ and discount $0.99$. All neural networks (each of the Multi-policy ones, the SNN and the Manager Network) have 2 layers of 32 hidden units. For the SNN training, the mesh density used to grid the $(x, y)$ space and give the MI bonus is 10 divisions/unit. The number of skills trained (ie dimension of latent variable in the SNN or number of independently trained policies in the Mulit-policy setup) is 6. The batch size and the maximum path length for the pre-train task are also the ones used in the benchmark (Duan et al., 2016): 50,000 and 500 respectively. For the downstream tasks, see Tab. 1.

| Parameter | Mazes | Food Gather |
|---|---|---|
| Batch size | 1M | 100k |
| Maximum path length | 10k | 5k |
| Switch time $\mathcal{T}$ | 500 | 10 |

Table 1: Parameters of algorithms for downstream tasks, measured in number of time-steps of the low level control

## B   RESULTS FOR GATHER WITH BENCHMARK SETTINGS

To fairly compare our methods to previous work on the downstream Gather environment, we report here our results in the exact settings used in Duan et al. (2016): maximum path length of 500 and batch-size of 50k. Our SNN hierarchical approach outperforms state-of-the-art intrinsic motivation results like VIME (Houthooft et al., 2016).

The baseline of having a Center of Mass speed intrinsic reward in the task happens to be stronger than expected. This is due to two factors. First, the task offers a not-so-sparse reward (green and red balls may lie quite close to the robot), which can be easily reached by an agent intrinsically motivated to move. Second, the limit of 500 steps makes the original task much simpler in the sense that the agent only needs to learn how to reach the nearest green ball, as it won't have time to reach anything else. Therefore there is no actual need of skills to re-orient or properly navigate the environment, making the hierarchy useless. This is why when increasing the time-horizon to 5,000 steps, the hierarchy shines much more, as can also be seen in the videos.

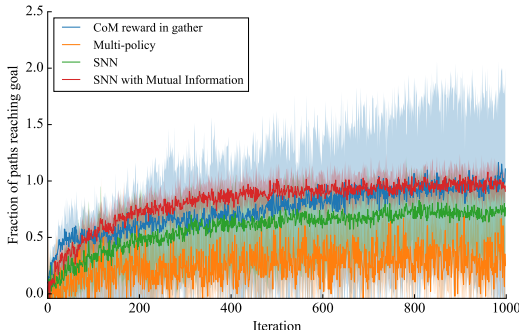

Figure 7: Results for Gather environment in the benchmark settings

## C   SNAKE

In this section we study how our method scales to a more complex robot. We repeat most experiments from Section 7 with the "Snake" agent, a 5-link robot depicted in Fig. 8. Compared to Swimmer, the action dimension doubles and the state-space increases by 50%. We also perform a further analysis on the relevance of the switch time $\mathcal{T}$ and end up by reporting the performance variation among different pretrained SNNs on the hierarchical tasks.

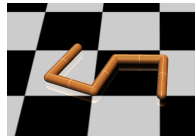

Figure 8: Snake

### C.1   SKILL LEARNING IN PRETRAIN

The Snake robot can learn a large span of skills as consistently as Swimmer. We report in Figs. 9(a)-9(e) the visitation plots obtained for five different random seeds in our SNN pretraining algorithm from Secs. 5.1-5.3, with Mutual Information bonus of 0.05. The impact of the different spans of skills on the later hierarchical training is discussed in Sec. C.4.

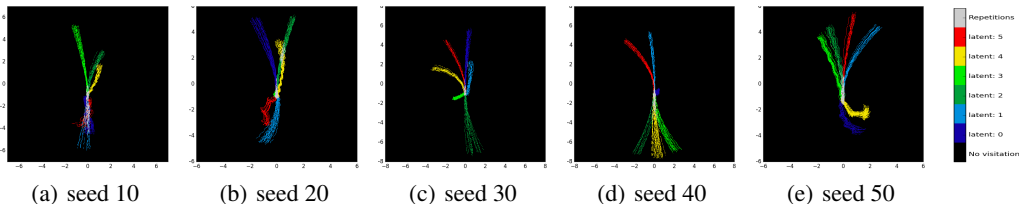

Figure 9: Span of skills learned using different random seeds to pretraining a SNN with MI = 0.05

## C.2 HIERARCHICAL USE OF SKILLS IN MAZE AND GATHER

Here we evaluate the performance of our hierarchical architecture to solve with Snake the sparse tasks Maze 0 and Gather from Sec. 6. Given the larger size of the robot, we have also increased the size of the Maze and the Gather task, from 2 to 7 and from 6 to 10 respectively. The maximum path length is also increased in the same proportion, but not the batch size nor the switch time $\mathcal{T}$ (see Sec. C.3 for an analysis on $\mathcal{T}$). Despite the tasks being harder, our approach still achieves good results, and Figs. 10(a)-10(b), clearly shows how it outperforms the baseline of having the intrinsic motivation of the Center of Mass in the hierarchical task.

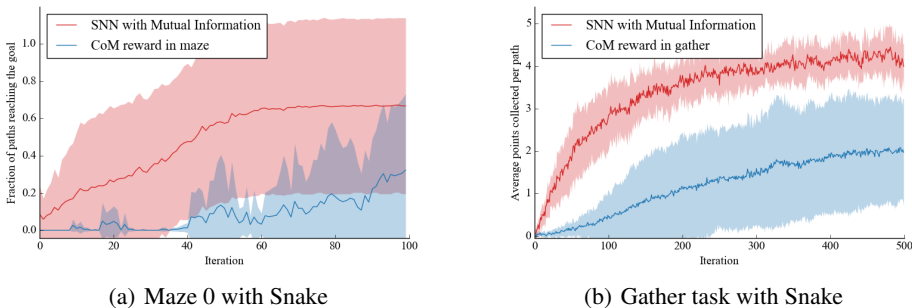

(a) Maze 0 with Snake (b) Gather task with Snake

Figure 10: Faster learning of our hierarchical architecture in sparse downstream MDPs with Snake

## C.3 ANALYSIS OF THE SWITCH TIME $\mathcal{T}$

The switch time $\mathcal{T}$ does not critically affect the performance for these static tasks, where the robot is not required to react fast and precisely to changes in the environment. The performance of very different $\mathcal{T}$ is reported for the Gather task of sizes 10 and 15 in Figs. 11(a)-11(b). A smaller environment implies having red/green balls closer to each other, and hence more precise navigation is expected to achieve higher scores. In our framework, lower switch time $\mathcal{T}$ allows faster changes between skills, therefore explaining the slightly better performance of $\mathcal{T} = 10$ or 50 over 100 for the Gather task of size 10 (Fig. 11(a)). On the other hand, larger environments mean more sparcity as the same number of balls is uniformly initialized in a wider space. In such case, longer commitment to every skill is expected to improve the exploration range. This explains the slower learning of $\mathcal{T} = 10$ in the Gather task of size 15 (Fig. 11(b)). It is still surprising the mild changes produced by such large variations in the switch time $\mathcal{T}$ for the Gather task.

For the Maze 0 task, Figs. 11(c)-11(d) show that the difference between different $\mathcal{T}$ is important but not critical. In particular, radical increases of $\mathcal{T}$ have little effect on the performance in this task because the robot will simply keep bumping against a wall until it switches latent code. This behavior is observed in the videos attached with the paper[2]. The large variances observed are due to the performance of the different pretrained SNN. This is further studied in Sec. C.4.

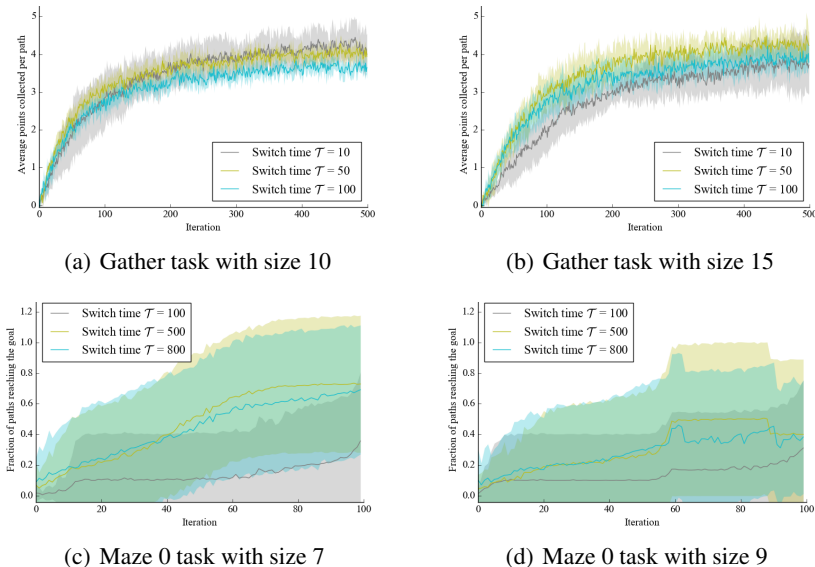

(a) Gather task with size 10 (b) Gather task with size 15

(c) Maze 0 task with size 7 (d) Maze 0 task with size 9

Figure 11: Mild effect of switch time $\mathcal{T}$ on different sizes of Gather and Maze 0

### C.4 IMPACT OF THE PRE-TRAINED SNN ON THE HIERARCHY

In the previous section, the low variance of the learning curves for Gather indicates that all pre-trained SNN preform equally good in this task. For Maze, the performance depends strongly on the pretrained SNN. For example we observe that the one obtained with the random seed 50 has a visitation with a weak backward span (Fig. 9(e)), which happens to be critical to solve the Maze 0 (as can be seen in the videos[2]). This explains the lack of learning in Fig. 12(a) for this particular pretrained SNN. Note that the large variability between different random seeds is even worse in the baseline of having the CoM reward in the Maze, as seen in Fig. 12(b). 3 out of 5 seeds never reach the goal and the ones that does have an unstable learning due to the long time-horizon.

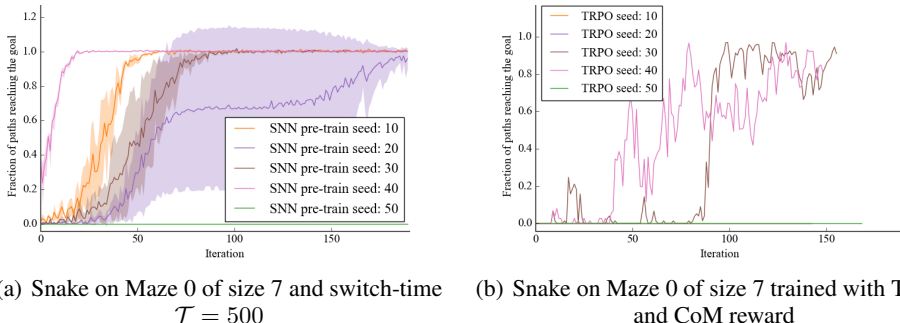

(a) Snake on Maze 0 of size 7 and switch-time (b) Snake on Maze 0 of size 7 trained with TRPO
$\mathcal{T} = 500$ and CoM reward

Figure 12: Variation in performance among different pretrained SNNs and different TRPO seeds

## D ANT

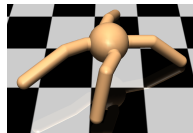

In this section we report the progress and failure modes of our approach with more challenging and unstable robots, like Ant (described in Duan et al. (2016)). This agent has a 27-dimensional space and an 8-dimensional action space. It is unstable in the sense that it might fall over and cannot recover. We show that despite being able to learn well differentiated skills with SNNs, switching between skills is a hard task because of the instability.

Figure 13: Ant

### D.1 SILL LEARNING IN PRETRAIN

Our SNN pretraining step is still able to yield large span of skills covering most directions, as long as the MI bonus is well tuned, as observed in Fig. 14. Videos comparing all the different gaits can be observed in the attached videos[6].

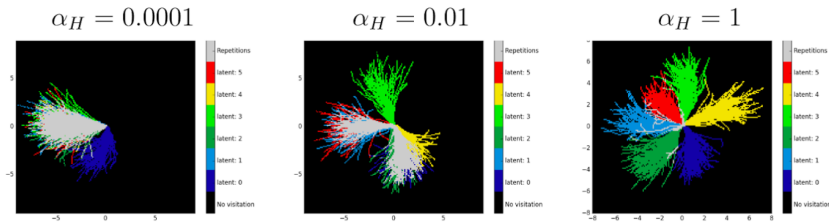

Figure 14: Pretrain visitation for Ant with different MI bonus

### D.2 FAILURE MODES FOR UNSTABLE ROBOTS

When switching to a new skill, the robot finds itself in a region of the state-space that might be unknown to the new skill. This increases the instability of the robot, and for example in the case of Ant this might leads to falling over and terminating the rollout. We see this in Fig. 15, where the 5 depicted rollouts terminate because of the ant falling over. For more details on failure cases with Ant and the possible solutions, please refer to the technical report: `http://bit.ly/snn4hrl-antReport`

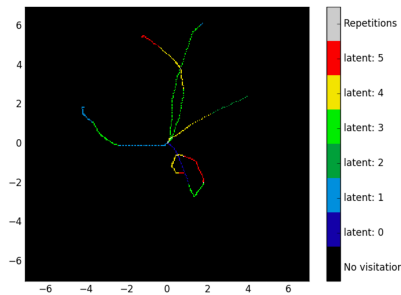

Figure 15: Failure mode of Ant: here 5 rollouts terminate in less than 6 skill switches.

---

[6]Videos available at: `http://bit.ly/snn4hrl-videos`

