# Peer review of "Stochastic Neural Networks for Hierarchical Reinforcement Learning"

_ICLR 2017 — accepted_

[Reviewer Comment · AnonReviewer1 · 12 Dec 2016]
**Clarification on experimental results**
clarity 4

Sorry for some last minute questions. What is the intuition for why the strong baseline in section 7.3 perform very poorly and can't solve the task? Assuming "CoM maze reward"is this baseline. Does the baseline also take into account the extra experience pre-trained policies have got? What is multi-policy, as compared to snn?

[Reviewer Comment · AnonReviewer1 · 12 Dec 2016]
**Snn architecture**
clarity 4

In figure 1, is it correct that (1) concatenation means based on categorical value, you have different bias to the first hidden layer, and (2) bilinear integration means you have different weight matrix connecting observation to the first hidden layer?

[Official Review · AnonReviewer2 · rating 8 · confidence 4 · 15 Dec 2016]
**No Title**
originality 4

I like the setting presented in this paper but I have several criticism/questions:

(1) What are the failure model of this work? As richness of behaviors get complex, I expected this approach to have issues with the diversity of skills that could be discovered.

(2) Looking at Sec 5.3 -- " let X be a random variable denoting the grid in which the agent is currently situated" -- is the space discretized? And if so why and what happens if it isn't. 

(3) Expanding on the first point, does the approach work with more complicated embodiment? Say a 5-link swimmer instead of 2? I think this is important to assess the generality of this approach

(4) Authors claim that "Recently, Heess et al. (2016) have independently proposed to learn a range of skills in a pre-training environment that will be useful for the downstream tasks, which is similar to our framework. However, their pre-training setup requires a set of goals to be specified. In comparison, we use intrinsic rewards as the only signal to the agent during the pre-training phase, the construction of which only requires very minimal domain knowledge."

I don't entirely agree with this. The rewards that this paper proposes are also quite hand-crafted and specific to a seemingly limited set of control tasks.

[Official Review · AnonReviewer3 · rating 7 · confidence 4 · 16 Dec 2016]

Interesting work on hierarchical control, similar to the work of Heess et al. 
Experiments are strong and manage to complete benchmarks that previous work could not. Analysis of the experiments is a bit on the weaker side.

(1) Like other reviewers, I find the use of the term ‘intrinsic’ motivation somewhat inappropriate (mostly because of its current meaning in RL).  Pre-training robots with locomotion by rewarding speed (or rewarding grasping for a manipulating arm) is very geared towards the tasks they will later accomplish. The pre-training tasks from Heess et al., while not identical, are similar. 

(2) The Mutual Information regularization is elegant and works generally well, but does not seem to help in the more complex mazes 1,2 and 3. The authors note this - is there any interpretation or analysis for this result?

(3) The factorization between S_agent and S_rest should be clearly detailed in the paper. Duan et al specify S_agent, but for replicability, S_rest should be clearly specified as well - did I miss it?

(4) It would be interesting to provide some analysis of the switching behavior of the agent. More generally, some further analysis of the policies (failure modes, effects of switching time on performance) would have been welcome.

[Official Review · AnonReviewer1 · rating 7 · confidence 4 · 16 Dec 2016 (modified: 25 Jan 2017)]
clarity 4

*Edited the score 6->7.

The paper presents a method for hierarchical RL using stochastic neural networks. The paper has introduced using information-theoretic measure of option identifiability as an additional reward for learning a diverse mixture of sub-policies. One nice result in the paper is the comparison with strong baseline which directly combines the intrinsic rewards with sparse rewards and shows that this supposedly smooth reward can’t solve tasks. Besides the argument made from the authors on difficulty on long-term credit assignment/benefits from hierarchical abstraction, one possible explanation for this might be the diversity requirement imposed in sub-policy training, which is assumed to be off in the baseline case. Wonder if this can shed insights into improving the baseline and proposing new end-to-end hierarchical policy learning as hierarchical REPS/option-critic etc. papers do. Nice visualizations.

The paper presents a promising direction, and it may be strengthened further by possibly addressing some of the following points. 

1) Limited diversification of sub-policies: Both concatenation and bilinear integration allow only minimal differentiations in sub-policies through first hidden weight, which is not a problem in the tested tasks because they essentially require same locomotion policies with minimal diversification, but such limitation can be more obvious in other tasks where ideal sub-policies are more diverse. Thus it is interesting to see it apply on harder, non-locomotion domains, where ideal sub-policies are not that similar, e.g. for manipulation, solving some task from one state can be very different from solving it from another state. 

2) Limitation on hierarchical policies: Manager network is trained while the sub-policies are fixed. Furthermore, the time steps for sub-policies are fixed. This requires “intrinsic” rewards and their learned sub-policies to be very good for solving down-stream tasks. It would be nice to see some more discussions/results on handling such cases, ideally connecting to end-to-end hierarchical policy learning.

3) Intrinsic/unsupervised rewards seem domain-specific/supervised rewards: Because of (2), this seems unavoidable.

[Author Response · Carlos Florensa · 24 Jan 2017]
**Updates summary**

Here we summarize the updates performed on the paper (uploaded):
         → Use term “Proxy reward” to refer to the reward in our pre-train environment.
         → Videos added on: skill diversity of “Ant”, and “Snake” solving Maze 0 and Gather. (see

[Final Decision · Program Chairs · 06 Feb 2017]
**ICLR committee final decision**

The authors provide an approach to policy learning skills via a stochastic neural network representation. The overall idea seems novel, and the reviewers are in agreement that the method provides an interesting take on skill learning. The additional visualization presented in response to the reviewer comments were also quite useful.
 
 Pros:
 + Conceptually nice approach to skill learning in RL
 + Strong results on benchmark tasks
 
 Cons:
 - The overall clarity of the paper could still be improved: the actual algorithmic section seems to present the methods still at a very high level, though code release may also help with this (it would not be at all trivial to re-implement the method just given the paper).
We hope the authors can address these outstanding issues in the camera ready version of their paper.